# The Correlation of Built Environment on Hypertension, and Weight Status amongst Adolescence in Saudi Arabia

**DOI:** 10.3390/ijerph192416763

**Published:** 2022-12-14

**Authors:** Anwar Al-Nuaim, Ayazullah Safi

**Affiliations:** 1Physical Education Department, Education College, King Faisal University, Al-Ahsa 31982, Saudi Arabia; 2Centre for Nutraceuticals, School of Life Sciences, University of Westminster, London W1W 6UW, UK

**Keywords:** youth, hypertension, sedentary behaviour, physical activity, health, environment

## Abstract

The prevalence of hypertension is becoming more common in children and adolescents than ever before. Thus, the aim of this study was to evaluate the associations between the built environment on physical activity, sedentary behaviour, waist circumference, and health amongst adolescents in Saudi Arabia. A systolic and diastolic blood pressure, resting heart rate and waist circumference of 380 boys and girls aged between 15–19 years old (male = 199 and females = 181) were measured. The International physical activity Questionnaire Short Form was used to assess the physical activity levels and time spent sitting. The statistical analysis conducted were means and standard deviation, 2-way and 3-way of variance (ANOVA), Bonferroni post hoc tests, Chi-squared distribution and Pearson’s correlations. Among males, 16.75% were classified as hypertensive, 12.69% as pre-hypertensive, and 70.56% as normal whereas, females, 23.20% were classified as hypertensive, 12.15% as pre-hypertensive and 64.64% as normal. There were significant differences (*F*_1,379_ = 16.50, *p <* 0.001) between males and females waist circumference. Pearson’s correlation also revealed significant positive relationships in sedentary time (*r =* 0.123, *p <* 0.016), WC (*r =* 0.104, *p <* 0.043), and systolic blood pressure (*r =* 0.110, *p <* 0.032). The results revealed that systolic and diastolic blood pressure are significantly related to multiple measures of weight status, and sedentary behaviour. The results also highlight that active youth had lower resting heart rate compared to inactive peers. The present findings provide a foundation of knowledge for future research and highlight the major need for research and policy interventions, to address the concerning health habits of Al-Ahsa youth and broader Kingdom of Saudi Arabia.

## 1. Introduction

Hypertension is one of the major public health concerns considered to be the most common risk factor for heart disease [1]. The prevalence of hypertension among children and adolescents has increased over the past few decades [2,3]. Hypertension tracks from childhood to adulthood [2,4], and high blood pressure during childhood and adolescent have been determined to be predictors of hypertension in adulthood [5,6]. According to the worldwide data, it was predicted that hypertension will increase to 1.56 billion people by 2025 globally [7]. In the Kingdom of Saudi Arabia, the prevalence of hypertension is an epidemic affecting 28.6% of males and 23.9% of females, and significantly higher among urban population compared to the rural population (27.9% and 22.4%, respectively) [8]. Several behavioural and environmental factors are linked with developing hypertension among children and adolescents [9]. Although several studies have found that obesity in adults is strongly associated with hypertension [10,11,12], yet few studies examined the association between obesity and hypertension during childhood and adolescence [13,14,15,16]. Moreover, physical activity (PA) is inversely associated with hypertension in adults [9,11,17,18,19,20].

Traditionally, studies have focused on individual demographics and psychological correlates to promote PA and prevent obesity [21,22,23]. The influence of cultural and environmental characteristics are expected to have a long-term and significant impact on the population, which could lead to the short-term effects of individually targeted interventions [24,25,26]. For instance, individuals from countries where PA is regarded as important for health are more likely to perceive fewer barriers, and therefore, engage in more regular PA compared to countries where PA is not a social norm [27]. Furthermore, cultural attitudes towards PA and environmental differences were postulated as explanations for British participants being more active compared to counterparts in Saudi Arabia [28]. Moreover, previous research has indicated that certain neighbourhood environmental characteristics such as residential density, accessibility, recreational facilities, aesthetics, and safety plays a major role in influencing PA and obesity [29]. Ref. [30] found that Indian urban women have higher mean values for both the waist and hip circumferences, five skinfold thicknesses as well as lean body mass, total fat and percentage fat compared to values for rural women. Relatively, Canadian children and adults from rural areas are more physically active than those living in the rural areas [31]. The number of nearby recreation facilities and parks were associated positively with girls’ PA engagement, whereas retail store ratio correlated positively with boys’ PA participation [32,33]. Ref. [34] found that children who live within one kilometre of a park playground were approximately five times more likely to be classified as a healthy weight compared to children without playgrounds in nearby parks. A systematic review [35] reported that neighbourhood residents who have access to supermarkets and limited access to convenience stores tend to have healthier diets and lower levels of obesity. Furthermore, [36] found that percentage of body fat and metabolic syndrome were significantly and inversely related to the increased distance to fast food and convenience stores. Perceived and objectively measured safety is a key element of the physical environment which can in part determine people’s PA behaviour [37,38,39]. Safety can be assessed either objectively (such as through assessing regional crime rates) or subjectively (such as through assessing people’s perceived safety in certain environments). Research has highlighted that both forms of assessment can be associated with young people’s PA levels, with a number of studies indicating that those who perceive their neighbourhood to be safe are more physically active and in turns leading healthy active lifestyle [40,41,42]. Although an increasing number of studies examining the association between PA levels and built environment has been documented, most studies were conducted in Western countries, particularly in the United State of America and the United Kingdom [43,44,45]. To the best of the researchers’ knowledge, no study conducted in Saudi Arabia aimed to determine the influences of built environment on PA and obesity. Thus, the aim of this study was to evaluate the association between built environment on hypertension, waist circumference and PA amongst adolescence in Saudi Arabia.

## 2. Methods

### Participants and Procedures

Following an institutional ethical approval an individual, parental, and school consents were gained prior commencing the data collection. A total of 380 secondary-school boys and girls aged 15–19 years old (male = 199; females = 181) representing different geographical areas such as urban, rural farm and rural desert of the Al-Ahsa Governorate participated in this study. 

## 3. Measures

### 3.1. Waist Circumference Measurement 

Direct measurements of waist circumference (WC) were obtained to the nearest 0.5 cm in accordance with the guidelines of [46]. The cut-off point for WC classification was a waist-to-height ratio of 0.5 [47]. Those who exceed 0.5 are considered “at risk” of cardiovascular diseases. WC is considered a simple measure of fat distribution in children and adolescents and is least affected by gender, race, and overall adiposity [47,48].

### 3.2. Resting Heart Rate and Blood Pressure Measures

Systolic blood pressure, diastolic blood pressure and resting heart rate were assessed using a digital portable device (BOSO-Medicus PC, Bosch and Sohn GmbH u. Co., Jungingen, Germany). Previous research has supported the use of BOSO-Medicus PC [49]. The blood pressure and resting heart rate were measured following the guidelines of National High Blood Pressure Education Programme [15]. Participants sat and rested in a quiet room for 5–10 min with their back supported and feet flat on the floor following the guidelines of the American Heart Association [50]. The blood pressure cuff was applied to the left arm with the lower margin of the cuff 2 centimetres (cm) above the elbow crease and with the arrow on the cuff aligned with the brachial artery. The cuff was wrapped to a tightness allowing one finger to be inserted under the top and bottom of the cuff. The average of two blood pressure readings were recorded at a two-minute interval. When the difference between the two first readings was greater than 2 millimetres of mercury (mmHg) for the systolic blood pressure measure, two additional measurements were taken after five minutes resting period, and the means of the last two measures were adopted until the differences did not exceed 2 mmHg. High blood pressure (the outcome variable) was defined as systolic blood pressure and/or diastolic blood pressure equal to or greater than the reference sex-, age-, and height-specific 95th percentile, following the NHBPEP criteria (Normal blood pressure: <90th percentile, pre-hypertensive blood pressure: ≥90th and <95th percentile and hypertensive blood pressure: ≥95th percentile [NHBPEP, 2004]). Hence, the three outcome variables were considered: high diastolic blood pressure, high systolic blood pressure, and high blood pressure.

### 3.3. Sedentary Behaviour-International PA Questionnaire Short Form (IPAQ-SF)

The IPAQ-SF was used to assess the time spent sitting. The IPAQ-SF is a 7-item scales, assessing the number of minutes spent in vigorous, moderate intensity activities, walking and time spent sitting in the previous seven days. Previous research has used the IPAQ and reported that it is the most valid and reliable tool for measuring PA levels and sitting time [51,52,53,54,55]. Therefore, in this study, the Arabic version of the IPAQ-E was used. For more information about the IPAQ-E version including translation process referred to [26].

## 4. Statistical Analysis

A range of statistical procedures were conducted to establish associations and differences in the under-researched population’s health and lifestyle habits from different locations and across genders such as urban, rural farm and rural dessert. The descriptive characteristics included mean and standard deviation (SD). Furthermore, comparisons were made between the lifestyle habits of youth from different weight status classification. Comparisons between genders, geographical locations, and age groups were analysed using the 2-way and 3-way of variance (ANOVA) on the participants sedentary behaviour, WC, systolic, and diastolic blood pressure. Moreover, Bonferroni post hoc tests were used to identify which groups were statistically significantly different. Prevalence rates according to gender and geographical locations were compared using Chi-squared distribution statistics. Furthermore, Pearson’s correlations were performed to establish relationships between health status variables (e.g., WC and blood pressure).

## 5. Results

### 5.1. Differences in Sedentary Behaviour 

Participants were asked how much time they spent sitting on a weekday. This included time spent sitting at a desk, visiting friends, reading, sitting, or lying down to watch television. The results showed no significant difference (*p >* 0.05) between males and females in sedentary time (3.14 and 2.96 h per day, respectively). In regard to geographical location, Univariate ANOVA revealed significant differences between participants in sedentary time (*p <* 0.015): participants from urban location spent more time sitting (3.29 h per day) than participants from rural farm location (2.95 h per day), but no significant difference was found between urban and rural desert, or between rural farm and rural desert (*p >* 0.05). Pearson’s correlation revealed significant positive relationships between sedentary time (*r =* 0.123, *p <* 0.016), WC (*r =* 0.104, *p <* 0.043), and systolic blood pressure (*r =* 0.110, *p <* 0.032). The descriptive statistics of mean and SD are outlined in Table 1.

### 5.2. Differences in Waist Circumference 

The univariate ANOVA showed a highly significant difference (*F*_1,379_
*=* 16.50, *p <* 0.001) between males and females in WC. Males (mean = 75.80 cm) had higher WC than females (mean = 70.38 cm). With reference to the WC gender-specific classifications, the percentages of “at-risk” males and females were 26.13% and 20.99%, respectively. 

### 5.3. Differences in Blood Pressure and Resting Heart Rate 

Univariate ANOVA revealed a significant difference between BP and WC (*F*_2,377_
*=* 10.48, *p <* 0.001), and WC (*F*_2,378_
*=* 5.01, *p =* 0.007). The Bonferroni post hoc pairwise comparisons revealed that there were significant differences between, WC of normal systolic blood pressure and hypertensive systolic blood pressure (22.68 and 25.99, and 71.60 cm and 76.27 cm, respectively). These are highlighted by the mean values in (Figure 1). Participants with normal WC reported to have significantly (*p <* 0.001) lower systolic blood pressure (118.03 mm Hg) compared to individual with higher (*p =* 0.005, 122.25 mm Hg). Additionally, participants with lower WC reported minimum health risks such as lower systolic blood pressure (*p <* 0.001) compared to those risk who reported higher (118.94 and 124.67 mm Hg). 

With respect to gender, univariate ANOVA showed a significant difference (*F*_1,377_
*=* 7.57, *p =* 0.006) between males and females in systolic blood pressure. Males (121.75 mm Hg) had higher systolic blood pressure than females (118.71 mm Hg). However, there was no significant difference between males (71.26 mm Hg) and females (70.72 mm Hg) in diastolic blood pressure (*p >* 0.05). Although males had significantly higher systolic blood pressure than females, Chi-square analysis revealed no significant gender difference (*p >* 0.05) regarding systolic blood pressure percentile interpretation. Among males, 16.75% were classified as hypertensive, 12.69% as pre-hypertensive, and 70.56% as normal. Among the females, 23.20% were classified as hypertensive, 12.15% as pre-hypertensive and 64.64% as normal. Moreover, Chi-square analysis revealed no significant difference (*p >* 0.05) between males and females regarding diastolic blood pressure percentile interpretation. In males, 4.06% were classified as hypertensive, 7.10% as pre-hypertensive and 80.83% as normal. Among the females 9.39% were classified as hypertensive, 6.63% as pre-hypertensive and 83.97% as normal. Furthermore, the percentage of youth who had either systolic or diastolic hypertension was 19.70% and 29.28% for males and females, respectively. Approximately 14% of males and 11% of females were pre-hypertensive and 66.16% of males and 59.67% had normal blood pressure. 

In addition, males had significantly (*F*_1,377_
*=* 50.72, *p <* 0.001) lower resting heart rate than females (80.43 and 90.71, respectively). The results showed no significant difference between males or females from different geographical locations (*p >* 0.05). Rural farm males had significantly (*p =* 0.022) lower systolic blood pressure than urban males (118.83 bpm and 123.73 bpm, respectively). Furthermore, univariate ANOVA revealed a significant difference (*F*_1,374_
*=* 28.87, *p <* 0.001) between active (75.6 bpm) youth and inactive youth (87 bpm) in resting heart rate. 

## 6. Associations between Health Risk Factors 

Pearson’s correlation revealed highly significant positive relationships between different measures of weight status (i.e., WC and waist height ratio (*p <* 0.001). In addition to the relation between weight with PA and blood pressure, both WC and waist height ratio had highly significant negative relationships with total METmin/wk (*r =* −0.169, *p =* 0.001 and *r =* −0.274, *p <* 0.001, respectively) and highly significant positive correlations with systolic blood pressure (*r* = 0.361, *p <* 0.001 and *r* = 0.310, *p <* 0.001, respectively). Moreover, total METmin/wk had highly significant negative correlations with systolic blood pressure (*r =* −0.146, *p =* 0.005) and resting heart rate (*r* = −0.271, *p <* 0.001), and a highly significant positive correlation with average steps per day (*r* = 0.370, *p <* 0.001). Finally, average steps per day had highly significant correlations with diastolic blood pressure (r = −0.151, *p =* 0.006) and resting heart rate (r = −0.219, *p <* 0.001) (see Table 2). In addition, Chi-square analysis revealed that active youth had a significantly (*χ*^2^_2_ = 8.944, *p =* 0.011) lower percentage of systolic or diastolic hypertension (9.09%); inactive youth had almost three times more as a percentage (26.48%).

Moreover, univariate ANOVA showed that active youth had significantly lower than inactive in WC (*p =* 0.045, 69.89 cm and 73.72 cm, respectively), waist height ratio (*p <* 0.001, 0.42 and 0.46, respectively), systolic blood pressure (*p <* 0.001, 116.93 and 120.79, respectively), and resting heart rate (*p <* 0.001, 75.60 and 87.00, respectively). However, there was no significant difference between PA in diastolic blood pressure, average steps per day (*p =* 0.012, 7940 and 6724 steps, respectively) and total METmin/wk (*p <* 0.001, 1455 and 616 METs, respectively). However, there was no significant difference between PA in diastolic blood pressure and resting heart rate (*p >* 0.05). Moreover, Chi-square analysis revealed that 18.22% of normal weight youth had significantly (*χ*^2^_2_ = 13.024, *p =* 0.001) higher percentage of systolic or diastolic hypertensive (18.22%) whereas overweight or obese had nearly double this percentage (34.27%).

## 7. Discussion

The rapid development and lifestyle changes in the Kingdom of Saudi Arabia, has contributed to increased sedentary behaviour, cardiovascular diseases, and hypertension in this society [56]. To our knowledge, this is the first study to be conducted in Saudi Arabia focusing on the impact of built environment on hypertension, sedentary behaviour, PA, health and weight status amongst adolescence in Saudi Arabia. While assessing the abdominal adiposity of youth by measuring WC, a significant gender difference was apparent; males had higher WC than females. The variation between genders in WC might be due to the difference between genders in the cut-off point of WC according to the National Institutes of Health guidelines (NHI). While the cut-off point for males is 102 cm, in females, the cut-off point is 88 cm. However, WC was measured as a predictor of obesity-related health risk [57], and the findings revealed that whilst males had higher WC there was no significant difference between the percentage of males (26.13%) and females (20.99%) classified as being at an increased health risk. The cut-off point for WC classification was a waist-to-height ratio of 0.5, with youth who exceed 0.5 being considered “at risk” of cardiovascular diseases [47].

The results of the current study revealed that both systolic and diastolic blood pressure are significantly related to multiple measures of weight status, and sedentary behaviour in Al-Ahsa youth. These results are consistent with a range of previous studies of children and young people that have indicated associations between blood pressure and both weight status [13,14,15,58,59] and PA levels [60,61]. For example, [62] found that BMI and WC were significantly associated with both systolic and diastolic blood pressure. Moreover, [13] reported that overweight adolescents had systolic and diastolic blood pressures that were 5.1 and 2.5 mmHg higher, respectively, compared with adolescents with a normal weight, while obese adolescents had blood pressure that was 11.3 and 6.2 mmHg higher compared to normal weight adolescents. The differences in the current study amount to an increase of 4.2 mmHg for SBP in overweight compared to normal weight youth and an increase of 8.5 mmHg for SBP in obese youth compared to normal weight youth. Ref. [60] reported that each additional hour per day of low light-intensity or MVPA could decrease the risk of having diastolic or systolic blood pressure. Moreover, [61] documented that PA could result in an increased production of nitric oxide; this has an effect on the regulation of blood pressure through endothelial vasodilatation. Accordingly, it can be inferred from this study’s findings that while they support previous research in the field, they do not constitute an original contribution.

The prevalence of hypertension was high among youth in Al-Ahsa. Females had a higher percentage of hypertensive systolic blood pressure (23.20%), and more than double of hypertensive diastolic blood pressure (9.39%) compared to males (16.75% and 4.06%, respectively). This prevalence of hypertension is higher compared with the findings from several previous international studies [15,16,19,63,64] In Brazil, [15] found that the prevalence of hypertension among adolescents was 10.5% in males and 9.9% in females. In another Brazilian study, [19] revealed that the prevalence of hypertension was higher amongst adolescent males (14.6%) compared with females (10.9%). In Chinese adolescents, [64] found 9.6% of males and 9.5% of males had hypertension, while [16] reported that 1.6% of Indian have systolic hypertension while 5.4% had diastolic hypertension. Finally, the prevalence of hypertension (systolic and/or diastolic blood pressure exceeded the 95th percentile) was 2.53% in Romanian adolescents in the age range 15–18 years [63]. This variation in prevalence of hypertensive adolescents might be due to weight status and adolescents’ and sedentary behaviour. Ref. [65] reported that the detection rates of hypertension differed depending on adolescents’ weight status. Moreover, [20] indicated that blood pressure was associated with decreased time in MVPA, as well as increased time in sedentary behaviour. Based on the current study findings and previous literature, it is clear that there is a concerningly high prevalence of hypertension within youth from Al-Ahsa governorate. Moreover, as suggested by previous literature, promoting PA, and tackling hypertension and obesity is fundamental in tackling this health issue.

Elevated resting heart rate in young people is predictive of later development of hypertension in adulthood [66]. Previous studies have indicated the link between elevated resting heart rate with other potential cardiovascular prognostic factors, including high blood pressure and obesity [67,68]. In the current study the results revealed that males had significantly lower resting heart rate than females. This could be a result of the differences between genders in PA levels. For instance, the current results highlighted that active youth had significantly lower resting heart rate compared to inactive peers. Moreover, [69] indicated that the resting heart rate difference between young people was identified between genders as well as PA level. These results supports previous research by [58] which reported that PA was negatively associated with higher resting heart rate. The current results are consistent with a range of previous studies in children and young people that have indicated associations between built environmental factors and both sedentary behaviour, and weight status [21,43,44,70,71,72].

## 8. Strength and Limitations

This study found its strength in evaluating the association between built environment on hypertension, waist circumference and PA amongst adolescences in Saudi Arabia. To the best of authors knowledge, this is the first study conducted in Saudi Arabia focusing on youth hypertension and health factors. The additional strength of this study was collecting data through various methods and the number of individuals participated in this study. This study contributes to the scarce literature and further highlight the necessity for future research and policy interventions, to address the concerning health habits of under research population. Despite the strength, this study is not without limitations. Limitations included participants from one region such as Al-Ahsa and targeted adolescents only. Collecting data from other regions and age groups may have yield different results. Additionally, qualitatively exploring reasons for sedentary behaviour and high WC may have provided an insight into why certain habits if any are followed.

## 9. Conclusions

This study provided an important insight into the influences of built environmental factors on Saudi youth health risk factors, including hypertension, sedentary behaviour, PA, WC, weight status and blood pressure. To the researcher’s knowledge, this is the first study in exploring the association between built environmental within youth in Saudi Arabia. Hence, the results of the current study support the generalizability of findings on the association between environmental factors across Western countries and Saudi Arabia. This study also provides a foundation of knowledge for future research to build upon and deliver interventions targeting sedentary behaviour and improving health amongst this group. Such findings will therefore further highlight the major need for research and policy interventions, to address the concerning health habits of Al-Ahsa youth and broader Kingdom of Saudi Arabia.

## Figures and Tables

**Figure 1 ijerph-19-16763-f001:**
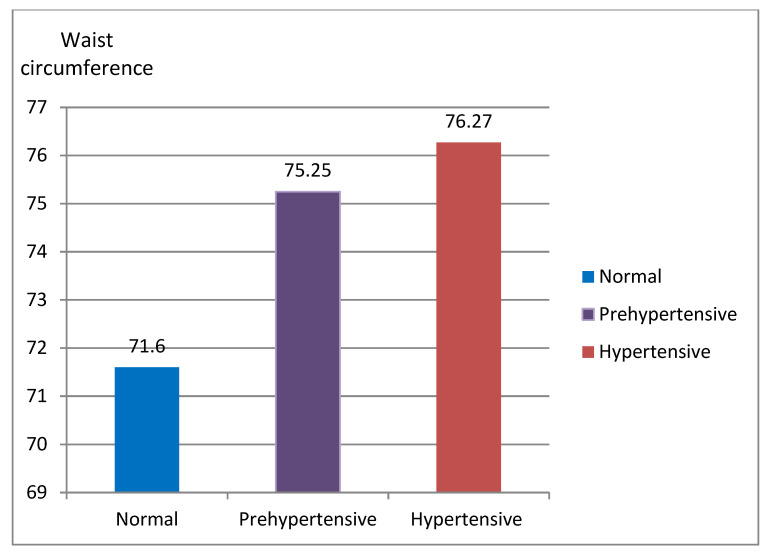
WC for youth across blood pressure classifications.

**Table 1 ijerph-19-16763-t001:** Mean *±* SD of the main dependent variables for the total sample and sub-samples.

Variable	Urban		Rural Farm		Rural Desert		Whole Sample		Total N = 380
	Male N = 96	Female N = 86	Male N = 54	Female N = 49	Male N = 49	Female N = 46	Male N = 199	Female N = 181	
**Age**	15.96 + 0.74	16.37 + 0.84	15.98 + 0.86	16.45 + 0.79	16.35 + 0.75	16.33 + 0.83	16.06 + 0.79	16.38 + 0.82	16.21 + 0.82
**Abdominal Circumference**	77.70 + 15.12	72.19 + 11.10	67.44 + 6.92	68.65 + 9.61	81.31 + 16.37	68.39 + 8.73	75.80 + 14.69	70.27 + 10.26	73.17 + 13.05
**Systolic blood pressure**	123.73 + 11.04	119.30 + 11.05	118.83 + 9.57	119.51 + 10.05	120.96 + 10.37	116.74 + 11.08	121.75 + 10.66	118.71 + 10.80	120.29 + 10.82
**Diastolic blood pressure**	75.35 + 59.32	71.77 + 7.93	67.54 + 7.83	69.35 + 7.80	67.18 + 11.27	70.24 + 7.43	71.26 + 42.06	70.72 + 7.80	71 + 30.80
**Resting heart rate**	80.54 + 12.70	89.33 + 14.17	82.81 + 14.77	94.18 + 13.82	77.31 + 13.44	89.61 + 16.57	80.34 + 13.53	90.71 + 14.80	85.30 + 15.06

**Table 2 ijerph-19-16763-t002:** Correlation coefficients of health risk factors variables.

		WC	WHR	Systolic	Diastolic	HR
WC	R value	1	0.944 **	0.361 **	0.013	−0.090
	*p* value		0.000	0.000	0.801	0.082
WHR	R value	0.944 **	1	0.310 **	0.027	0.001
	*p* value	0.000		0.000	0.607	0.984
Systolic	R value	0.361 **	0.310 **	1	0.098	0.110 *
	*p* value	0.000	0.000		0.058	0.032
Diastolic	R value	0.013	0.027	0.098	1	0.089
	*p* value	0.801	0.607	0.058		0.084
HR	R value	−0.090	0.001	0.110 *	0.089	1
	*p* value	0.082	0.984	0.032	0.084	

** Correlation is significant at the 0.01 level. * Correlation is significant at the 0.05 level.

## Data Availability

Not applicable.

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
