# Peer review of "The Correlation of Built Environment on Hypertension, and Weight Status amongst Adolescence in Saudi Arabia"

_ijerph, 2022, doi:10.3390/ijerph192416763_

Round 1
Reviewer 1 Report
This is a very comprehensive study looking at a range of different variables relating to health status of Saudi citizens. This indeed is a country which has had few studies of this kind, and credit goes to the authors for carrying out such an exhaustive investigation. The study is supported with a wide range of literature, and it draws excellent comparisons with other nation's findings.
There are a few minor corrections which I would recommend before accepting before publication. Please see below:
1. Abstract: Spelling error – ‘Discission and Conclusion’
2. Line 38 – ‘United Kingdom of Saudi Arabia’ – Should this be ‘Kingdom of Saudi Arabia’?
3. Line 68 – ‘review by’ – This needs rewording (‘by’ should be deleted)
4. Figure 1 – Y Axis needs labelling
5. Line 274 – Inconsistencies in font size
6. Line 282 – Inconsistencies in font size
7. Line 287 – ‘resting heart rate different’ – should this read ‘resting heart rate difference’.
The methodological design of this study incorporates a cross-sectional approach providing an indication of current health behaviours of the Saudi population at a given time of year. However, did the authors give any consideration to adopting a more longitudinal research design? This would provide a more reflective indication of health behaviours, and would consequently provide more meaningful conclusions. Additionally, could the authors then draw comparisons to additional variables which may influence Saudi health across the year? i.e. seasonal variation, the holy month of Ramadan, public holidays etc.
The study is supported with sufficient data which has been analysed coherently to produce valid conclusions. Did the authors consider whether there was any associations between the variables? i.e. would it be worth future studies exploring the association between blood pressure and the IPAQ? Additionally, did the authors analyse the gender data according to age? Did this provide any findings of interest for any of the variables?
Author Response
There are a few minor corrections which I would recommend before accepting before publication. Please see below:
- Abstract: Spelling error – ‘Discission and Conclusion’
Author response: Thank you for your comment. This has been corrected
- Line 38 – ‘United Kingdom of Saudi Arabia’ – Should this be ‘Kingdom of Saudi Arabia’?
Author response: Thank you for your comment. This has been corrected to the ‘Kingdom of Saudi Arabia’
- Line 68 – ‘review by’ – This needs rewording (‘by’ should be deleted)
Author response: Thank you for your comment. This has been corrected and “by” has been deleted.
- Figure 1 – Y Axis needs labelling
Author response: Thank you for your comment. Figure 1 – Y axis label has been added as “Waist circumference”
- Line 274 – Inconsistencies in font size
Author response: Thank you for your comment. This has been corrected
- Line 282 – Inconsistencies in font size
Author response: Thank you for your comment. This has been corrected.
- Line 287 – ‘resting heart rate different’ – should this read ‘resting heart rate difference’.
Author response: Thank you for your comment. This has been corrected
The methodological design of this study incorporates a cross-sectional approach providing an indication of current health behaviours of the Saudi population at a given time of year. However, did the authors give any consideration to adopting a more longitudinal research design? This would provide a more reflective indication of health behaviours and would consequently provide more meaningful conclusions. Additionally, could the authors then draw comparisons to additional variables which may influence Saudi health across the year? i.e., seasonal variation, the holy month of Ramadan, public holidays etc.
Author response: Thank you for your comment and suggestions. We think the current paper will open doors for more longitudinal research and that is something we are considering doing. In regards to the seasonal variations and Holy month of Ramadan and public holidays are also great suggestions and plan for the future studies.
The study is supported with sufficient data which has been analysed coherently to produce valid conclusions. Did the authors consider whether there was any associations between the variables? i.e., would it be worth future studies exploring the association between blood pressure and the IPAQ? Additionally, did the authors analyse the gender data according to age? Did this provide any findings of interest for any of the variables?
Author response: Thank you for your comment and suggestions. We think there is potential to investigate/explore the association between IPAQ and blood pressure and future studies should look into this. We have provided the age and gender results in table 1 and have communicated the interest of these variables in results and discussion sections.

Reviewer 2 Report
1. The research method does not explain the relationship between waist circumference and research purpose, which is too one-sided. For example, there is no obvious relationship between waist circumference and health problems such as hypertension.
2. Most of the articles focus on the study of hypertension, waist circumference and sedentary time, however the relationship with the building environment is missed. For example, the discussion mentioned the comparison of the relationship and comparison of waist circumference, blood pressure, weight and other physical indicators, but did not mention the impact of the building environment on these values.
3. It mentioned the prevalence of hypertension among adolescents in other countries, which did not help the study much, it seems just mentioned briefly.
4. The difference between rural and urban areas is shown in the survey of sedentary time, and the comparison of heart rate between rural men and urban men is mentioned in the resting heart rate, which is not enough to support the research results.
5. In statistics analysis, it is mentioned that research on the association and difference of health and living habits among people of different regions and genders, such as adolescents in urban, rural and desert areas. However, in the follow-up research, there is no obvious difference between different populations or genders, and no comparison of research objects and environment is made.
6. The number of daily walks mentioned in associations between health risks factors can be used as a research on the relationship with the building environment, but this research is not mentioned in the article.
Author Response
Reviewer 2:
- The research method does not explain the relationship between waist circumference and research purpose, which is too one-sided. For example, there is no obvious relationship between waist circumference and health problems such as hypertension.
Author response: Thank you for your comment and suggestions. We are not sure what exactly is asked here to consider but the information about waist circumference and risk to health are outlined in section “measures” line 95-101 and we used the existing research to support the waist circumference and health problem. If there is anything that we have not covered and could be suggested we may consider looking into it.
- Most of the articles focus on the study of hypertension, waist circumference and sedentary time, however the relationship with the building environment is missed. For example, the discussion mentioned the comparison of the relationship and comparison of waist circumference, blood pressure, weight, and other physical indicators, but did not mention the impact of the building environment on these values.
Author response: Thank you for your comment and suggestions. We have provided information about the build environment and also to our knowledge this is the first study to look into the influences of built environment in Saudi. “Thus, the aim of this study was to evaluate the association between built environment on hypertension, waist circumference and PA amongst adolescence in Saudi Arabia”. Furthermore, in the discussion section starting from line 230 “we outlined with support of existing research that “the rapid development and lifestyle changes in the Kingdom of Saudi Arabia, has contributed to increased sedentary behaviour, cardiovascular diseases, and hypertension in this society” (56). We also, aligned our findings with previous research conducted in various developed countries as outlined in line 266 onwards where we discussed the countries and gender differences. Furthermore, we also discussed and aligned to existing research at the last paragraph of our discussion to the impact of built environment from line 295 onwards. Also, the impact of build environment was mentioned in the conclusion section as well. If there is anything that we have not covered and could be suggested we may consider looking into it.
- It mentioned the prevalence of hypertension among adolescents in other countries, which did not help the study much, it seems just mentioned briefly.
Author response: Thank you for your comment and suggestions. We are not sure what exactly is requested to produce regarding this but when we mentioned ‘hypertension among adolescents in other countries’ we provided supportive/compared/contrasted our findings with the existing research in particular from line 264 -278. If there is anything that we have not covered and could be suggested we may consider looking into it.
- The difference between rural and urban areas is shown in the survey of sedentary time, and the comparison of heart rate between rural men and urban men is mentioned in the resting heart rate, which is not enough to support the research results.
Author response: Thank you for your comment and suggestions. We are not sure what exactly is requested to produce regarding the results, but we believe various relevant tests and analysis have been conducted and the results produced and reported sufficiently supports the research results. The results section is from line 145 – 229 that clearly outlined all the necessary tests and sufficient results. If there is anything that we have not covered and could be suggested we may consider looking into it.
- In statistics analysis, it is mentioned that research on the association and difference of health and living habits among people of different regions and genders, such as adolescents in urban, rural, and desert areas. However, in the follow-up research, there is no obvious difference between different populations or genders, and no comparison of research objects and environment is made.
Author response: Thank you for your comment and suggestions. We are not sure what exactly is requested to further produce about this, but we believe we produced and provided sufficient comparison of research object and environment. We produced and provided descriptive statistics that outlines the comparison or differences between urban, rural farm, rural desert, and whole sample in table 1. We also provided the comparison for sedentary behaviour and within all the other results. Please read the result section from line 145 – 229. If there is anything that we have not covered and could be suggested we may consider looking into it.
- The number of daily walks mentioned in associations between health risks factors can be used as a research on the relationship with the building environment, but this research is not mentioned in the article.
Author response: Thank you for your comment and suggestions. We are not sure what exactly is requested to further produce about this, but we believe this has been mentioned in the article in the results section line 202 – 228 and also discussed on the discussion section. If there is anything that we have not covered and could be suggested we may consider looking into it.

Reviewer 3 Report
Thank you for the opportunity to review such an interesting work. Admittedly, the authors made many mistakes, but I believe that after applying corrections, the work will be worth publishing in the journal.
Major points:
- The authors gave too general the analytical procedures used. Can it be expanded?
- Please explain what are the strengths and limitations of your study?
Minor points:
-Abstract – is too long, and please remove headings
- References list – please adjust the list to the requirements of the journal and add where it is possible doi number
- References list is too extensive, this is not a review but a research work, please delete the least important items
- Please check and sign all abbreviations used in the tables.
- Figure 1 - please sign the y axis
- a Native speaker proofreading is recommended
After all corrections have been made, the manuscript should be published.
Author Response
Reviewer 3
Comments and Suggestions for Authors
Thank you for the opportunity to review such an interesting work. Admittedly, the authors made many mistakes, but I believe that after applying corrections, the work will be worth publishing in the journal.
Major points:
- The authors gave too general the analytical procedures used. Can it be expanded?
Author response: Thank you for your comment and suggestions. We are not sure what exactly is asked here but the reason a general analytical procedure is used because of the nature of the research and aims of this study. We have followed up studies and depending on the aims the analytical procedures could be extended in the future follow up articles. However, meanwhile If there is anything that we have not covered and could be suggested we may consider looking into it.
- Please explain what are the strengths and limitations of your study?
Author response: Thank you for your comment and suggestions. A section about strength and limitations have been added from line 301-314.
Minor points:
-Abstract – is too long, and please remove headings
Author response: Thank you for your comment and suggestions. The abstract has been condensed and heading has been removed.
- References list – please adjust the list to the requirements of the journal and add where it is possible Doi number.
Author response: Thank you for your comment and suggestions. This has been amended.
- References list is too extensive, this is not a review but a research work, please delete the least important items.
Author response: Thank you for your comment and suggestions. We understand the extensive reference list but feel there is a need for them. Therefore, no change made to the number of references list.
- Please check and sign all abbreviations used in the tables.
Author response: Thank you for your comment and suggestions. These has been amended
- Figure 1 - please sign the y axis
Author response: Thank you for your comment. Figure 1 – Y axis label has been added as “Waist circumference”
- a Native speaker proofreading is recommended.
Author response: Thank you for your comment. The article has been proofread by a native speaker and approved.
After all corrections have been made, the manuscript should be published
